# SMC5/6: Multifunctional Player in Replication

**DOI:** 10.3390/genes10010007

**Published:** 2018-12-22

**Authors:** Jan J. Palecek

**Affiliations:** 1National Centre for Biomolecular Research, Faculty of Science, Masaryk University, Kotlarska 2, 61137 Brno, Czech Republic; jpalecek@sci.muni.cz; Tel.: +420-54949-6128; 2Mendel Centre for Plant Genomics and Proteomics, Central European Institute of Technology, Masaryk University, Kamenice 5, 62500 Brno, Czech Republic

**Keywords:** SMC complexes, SMC5/6, chromatin structure, loop extrusion, stalled replication fork, replication fork regression, collapsed replication fork, homologous recombination

## Abstract

The genome replication process is challenged at many levels. Replication must proceed through different problematic sites and obstacles, some of which can pause or even reverse the replication fork (RF). In addition, replication of DNA within chromosomes must deal with their topological constraints and spatial organization. One of the most important factors organizing DNA into higher-order structures are Structural Maintenance of Chromosome (SMC) complexes. In prokaryotes, SMC complexes ensure proper chromosomal partitioning during replication. In eukaryotes, cohesin and SMC5/6 complexes assist in replication. Interestingly, the SMC5/6 complexes seem to be involved in replication in many ways. They stabilize stalled RFs, restrain RF regression, participate in the restart of collapsed RFs, and buffer topological constraints during RF progression. In this (mini) review, I present an overview of these replication-related functions of SMC5/6.

## 1. Introduction

Replication in any organism is constantly challenged at many levels. A deficit (or inhibition) in the deoxynucleoside triphosphate (dNTP) pool can slow or pause the replication fork (RF), damaged or modified DNA can block RF progression (and must be either repaired first or trans-replicated through the damaged site), some highly transcribed sequences must be replicated in the same direction as transcription (to avoid head-on collisions), the RF must “pause” at certain sites (such as repetitive and highly transcribed ribosomal DNA (rDNA) sequences), or a stalled RF can even collapse and must be restarted. Furthermore, the genome replication involves not only a physical duplication of DNA molecules, but also its spatial organization, which must be tightly coupled with the duplication process. DNA must be specifically packed to fit into the nuclear space, especially when the DNA mass is doubled during its duplication. One of the most important factors organizing DNA into higher-ordered structures are Structural Maintenance of Chromosome (SMC) complexes.

In prokaryotes, the SMC/ScpAB complexes (Scp stands for segregation and condensation protein) play key roles in the bacterial chromosome partitioning during replication and the successful segregation of the bacterial nucleoid [1,2]. The eukaryotic SMC complexes have distinct functions in the organization of chromatin to higher-ordered structures. Cohesins hold newly replicated sister chromatids together, promote sister chromatid recombination, and are responsible for the dynamic organization of chromatin fibers during interphase into topologically associated domains (TADs). Condensins play key roles in the compaction and individualization of chromatids in mitosis. The SMC5/6 complexes are implicated in the repair of DNA damage by homologous recombination, stabilization and restarting of stressed RFs, and the resolution of DNA superhelical tension that is induced by DNA replication [3,4,5,6,7,8,9,10,11,12,13,14].

All SMC complexes are primarily built of long coiled-coil SMC proteins that use ATP hydrolysis to drive changes in DNA topology. The SMC proteins make dimers via their hinge domains (Figure 1) [15,16,17,18,19]. In addition to the constitutive dimerization via their hinge domains, two SMC molecules transiently interact when their head domains sandwich a pair of ATP molecules. The ATPase head domains are also bridged by a kleisin subunit (Nse4 in SMC5/6), which binds the coiled-coil base region immediately adjacent to one SMC (designated as ν-SMC; SMC6 subunit in SMC5/6) head domain via its amino terminus [20,21,22] and to the distal side of the other SMC (designated as κ-SMC; SMC5 subunit in SMC5/6) head domain via its carboxyl terminus (Figure 1) [20,23,24]. This arrangement imposes overall asymmetry even upon prokaryotic SMC homodimers, and results in a closed ring structure, which is able to encircle DNA within its circumference [21,25,26,27,28].

In addition, all kleisin proteins recruit supplementary protein subunits that are either composed of HEAT repeat motif (Hawk; HEAT proteins associated with kleisin) proteins, as in the case of cohesin and condensin, or tandem winged helix domain (WHD) (Kite; kleisin-interacting tandem winged-helix element) proteins, as in the case of prokaryotic SMC/ScpAB and eukaryotic SMC5/6 complexes (Figure 1; Nse1 and Nse3 subunits in SMC5/6) [29,30,31]. Kite and Hawk subunits participate in kleisin bridge regulation and binding to DNA [32,33,34]. The similar core composition and architecture of the eukaryotic SMC5/6 and prokaryotic SMC complexes led us recently to propose their close evolutionary and functional relationship [29]. Particularly, both complexes are involved in replication processes. However, the SMC5/6 complexes acquired new complexity (as eukaryotes did) during their evolution. For example, the Nse1 Kite subunit acquired a novel ubiquitin-ligase domain [35,36] and a new unique conserved Nse2/Mms21 subunit is positioned at the arm of SMC5 and it possesses small ubiquitin-like modifier (SUMO)-ligase activity (Figure 1) [37,38,39,40,41,42,43,44,45,46]. In addition, weakly conserved Nse5 and Nse6 subunits were identified in yeast, plant, and mammalian organisms [47,48,49,50]. Positions of Nse5 and Nse6 on the SMC5–SMC6 dimer differ in *Saccharomyces cerevisiae* and *Schizosaccharomyces pombe* organisms [24,51]. Functions of these evolutionary divergent subunits seem to be related to SUMOylation and ubiquitination activities of the complex.

The circular shape and the size of the SMC rings allow them to encircle two DNA strands. It was demonstrated that the SMC complexes can entrap DNA in a topological way (Figure 2a) [52,53,54]. For example, this DNA-binding mode enables cohesin rings to hold newly replicated sister chromatids until the onset of anaphase, at which point they are released by cleavage of kleisin and allowed to segregate [55]. In addition, ATP binding and hydrolysis provide the SMC complexes with motor activity that allows them to slide along the entrapped DNA molecule(s). This SMC translocation activity may progressively enlarge loops of DNA in a process termed loop extrusion (Figure 2b) [2,6,56,57,58,59,60,61,62,63]. For example, the condensin and SMC/ScpAB complexes translocate along the DNA strands leaving loop(s) behind [59,64,65].

In this (mini) review, I first briefly summarize the SMC5/6 data related to the genome integrity, and then I focus on the SMC5/6 role(s) in the normal progression of replication.

## 2. SMC5/6 Roles in Maintenance of Genome Integrity

Thanks to numerous studies, SMC5/6 was implicated in homologous recombination, stabilization of stalled RFs, restarting of collapsed RFs, maintenance of ribosomal DNA (rDNA) and heterochromatin, telomerase-independent telomeres elongation, and the regulation of chromosomal topology (see below and other reviews [3,11,12]). The nature of these functions implies that the SMC5/6 complex also contributes to meiotic processes, including meiotic recombination [68].

### 2.1. SMC5/6 in Homologous Recombination

The first *smc6* mutation (originally called a *rad18-X* mutant) was identified in the yeast *S. pombe* where it conferred hypersensitivity to ultraviolet (UV), infrared (IR), and chemical agents causing DNA damage or affecting replication [69]. The *smc6* mutations are genetically epistatic with *S. pombe* rad51/rhp51, indicating a function in homologous recombination (HR) downstream of rad51/rhp51 [47,70,71]. In the yeast *S. cerevisiae*, scSMC5/6 and cohesin are recruited to double-strand breaks (DSBs) in a *RAD50/MRE11*-dependent manner in gap 2/mitosis (G2/M) when sister chromatids are present and HR is the major DNA repair pathway (Figure 3a) [72,73]. In humans, the recruitment of *Homo sapiens* (hs)SMC5/6 seems to be dependent on RAD18 and SLF1/SLF2 dimer (a distant ortholog of the yeast Nse5/Nse6 dimer; [48,74]). It was shown that SMC5/6 promotes sister chromatid recombination (SCR) in many organisms, such as yeast [72], human [43,74], *Arabidopsis* [75,76], and chicken [77]. SMC5/6 and cohesin work in the same pathway, as suggested by the epistasis analysis of their mutants [74,78]. Like cohesin, the SMC5/6 complex may hold two DNA strands inside its ring and align them to promote HR [12]. However, apart from such a direct role of SMC5/6 in HR, the SMC5/6 enzymatic activities might be (at least partly) responsible for SCR promotion via cohesin modifications (Figure 3a). In human and *S. cerevisiae*, the SMC5/6 subunit Nse2/Mms21 SUMOylates the Scc1 subunit of cohesin and promotes SCR [38,43,79].

Yet, SMC5/6 may also regulate HR negatively—at repetitive sequences—as HR at repetitive sequences must be tightly controlled to avoid loss of genetic information. SMC5/6 is enriched in repetitive regions (such as rDNA, transfer DNA (tDNA), centromeres, and telomeres [72,73,80,81,82,83,84,85]) and plays both direct and indirect roles at these loci (Figure 3b). Again, SUMOylation of the cohesin (and condensin) subunits plays an important role in the maintenance of rDNA repeats, as the rDNA copy number and segregation are altered in the *nse2/mms21* SUMO-deficient mutant [46]. Directly, SMC5/6 reduces HR between repeats by promoting transport of the DSBs away from the surrounding repeats [86,87,88,89]. In insects and fungi, SMC5/6 binds heterochromatin protein 1 (HP1) and blocks HR at heterochromatin (containing repeats). DSBs are then located outside heterochromatin, where subsequent steps in HR take place (without the danger of ectopic recombination between repeats). Nse2-mediated SUMOylation triggers the relocalization of repair sites to nuclear periphery, where SMC5/6 interacts with the SUMO-targeted ubiquitin-ligase (STUbL/RENi) complex, which ubiquitylates SUMOylated proteins and enables HR progression (Figure 3b) [89,90,91]. Note, that the SMC5/6 (Nse2 SUMO-ligase activity) role in organizing nuclear movement is also reflected by its importance in clustering telomeres to the nuclear envelope in *S. cerevisiae* or their recruitment to ALT-associated promyelocyctic leukemia (PML) bodies in humans (due to the SUMOylation of telomeric proteins) [40,44,92]. Nevertheless, not all the defects at repetitive loci in the *smc5/6* mutants can be ascribed to aberrant HR, as the segregation defects at rDNA seem to be HR-independent and relocalization does not suppress them; instead, incomplete replication of rDNA causes segregation defects (see below) [93].

### 2.2. SMC5/6 Role at Stressed Replication Forks

In addition to epistatic interactions with the *rad51* deletion, *smc5/6* hypomorphic mutants are synthetically lethal with mutations in genes promoting recovery from stalled replication (such as srs2, sgs1/rqh1, and mus81), suggesting a role for SMC5/6 in recovery of stalled RFs by recombination [72,80,94,95,96]. Also, when RF collapses and the replisome is unloaded, HR is needed for its restart. SMC5/6 localizes to collapsed RF in *S. cerevisiae* and its mutants show arrest at the state of X-shaped joint molecules, suggesting a problem with their resolution (Figure 3c) [73,80,81,84,95,96]. Synthetic lethality of *mus81* or *sgs1/rqh1*, which process recombination junctions, further suggests a role for SMC5/6 in dealing with such intermediates. Increasing the ability of cells to resolve joint molecules can suppress formation of X-shaped recombination structures and revert *smc5/6* segregation phenotypes [47,71,97]. SMC5/6 may play both a structural and regulatory role, as the Sgs1/Top3/Rmi1 (STR) helicase complex is recruited and SUMOylated by SMC5/6 [98,99,100,101]. Firstly, auto-SUMOylation of the SMC5/6 subunits leads to recruitment of Sgs1 (mediated by two SUMO-interacting motifs on Sgs1) and, secondly, SMC5/6-dependent SUMOylation of STR promotes its helicase function. In conclusion, the SMC5/6 and its Nse2/Mms21 SUMO-ligase are involved in the dissolution of recombination intermediates that block replication completion (Figure 3a,c) [98,100].

However, in *S. cerevisiae*, scSMC5/6 also interacts with and regulates the scMph1 helicase. Interestingly, inactivation of scMph1 suppresses *smc5/6* mutants, suggesting that scSMC5/6 prevents the accumulation of toxic recombination intermediates generated by scMph1 [102,103,104,105]. Zhao and her colleagues showed that scSMC5/6 restrains replication fork regression activity of the Mph1 helicase (Figure 3c), but not its D-loop disruptive activity [106,107]. Further analysis suggested that rDNA repeats, containing tens to hundreds of replication-fork-pausing sites (RPS), are the most prone to fork regression (particularly after inactivation of SMC5/6 [108]) and that reducing fork pausing (similar to *mph1* deletion) improves rDNA replication in cells without SMC5/6. At highly transcribed rDNA, the RNA–DNA hybrids can block RF progression, and Mph1 must be controlled by SMC5/6 to avoid toxic RF regression [109,110]. In contrast to STR helicase, Mph1 regulation does not seem to be SUMO-dependent, suggesting a sole mechanistic restriction posed by the SMC5/6 complex (Figure 3c) [102,106]. While an STR-related function is committed later during HR, the Mph1-related restraining role prevents HR and stabilizes stalled (paused) RFs [111].

Another piece of data also points to natural RPSs, as SMC5/6 is enriched at RPSs (rDNA, tDNA, centromeres, and telomeres) and co-localizes with Rrm3 helicase, which facilitates fork passage through these sites [84,112,113]. The sickness/lethality of the *smc6 rrm3∆* double mutant is rescued by individual deletions of Tof1-Csm3 (a fork protection complex that enforces pausing at RPS). This indicates that prolonged pausing and unfinished replication at rDNA, which represents 10% of the yeast genome, could be the major contributor to the observed phenotypes in *S. cerevisiae* [41,93]. Generally, *smc5/6* mutations, causing combined defects in DNA damage tolerance and pausing site replication, lead to recombination-mediated DNA lesions that are, however, invisible to the checkpoints. This results in premature mitosis with interconnected chromatids (that are cut during cytokinesis) and chromosome breakages [99,114,115].

## 3. SMC5/6 Is Involved in Normal Progression of Replication

The SMC5/6 functions described above become indispensable when cells are exposed to genotoxic stresses or when the replication process is challenged at specific sites (such as repetitive rDNA). However, localization and phenotypic data suggest that SMC5/6 may also play a role during unchallenged replication.

In *S. cerevisiae*, the scSMC5/6 complex is present on chromosomes at sites of replication initiation (autosomal replication sites (ARS)) during the synthesis (S) phase [73,116,117]. In G2/M, ARS localization is reduced and scSMC5/6 is distributed in a manner reminiscent of cohesin. This localization requires sister chromatid cohesion in *S. cerevisiae* [73]. In *S. pombe*, spSMC5/6 sites also significantly overlap with cohesin distribution, and spSMC5/6 is required for timely removal of cohesin from chromosome arms, suggesting the intimate relationship of these two SMC complexes for chromosomal structure maintenance [118,119,120].

A change in scSMC5/6 localization from ARS to cohesin-like distribution during cell cycle may indicate that the complex is associated with the fork and follows fork progression (and then it dissociates during mitosis). Furthermore, scSMC5/6 binding along chromosome arms is enhanced in a replication-dependent manner after Top2 inactivation. This increased chromosomal association of scSMC5/6 is independent of recombination, DNA breaks, and replication fork stalling (and no increase in X-shaped structures is observed [121]), suggesting that this is not induced by DNA damage or stressed RFs. Rather, enhanced chromosomal association of scSMC5/6 correlates with a segregation-inhibiting structure that can be removed by Top2 reactivation after the completion of replication (but persists during a prolonged G2/M arrest when Top2 is not reactivated). These data suggest that the scSMC5/6 chromosomal association may correlate with increased superhelical tension and the formation of sister chromatid intertwines (SCIs) during replication (Figure 3d) [121,122,123,124]. Sjogren and her colleagues proposed that SMC5/6 organizes sister chromatids in a way that promotes fork rotation [11,121]. They also showed that purified scSMC5/6 promotes Top2-dependent catenation of plasmids, suggesting that SMC5/6 interconnects two DNA molecules using ATP-regulated topological entrapment of DNA, similar to cohesin [66]. However, sedimentation and electrophoretic assays using yeast extracts containing special minichromosomes suggested no significant role of scSMC5/6 in minichromosome catenation (persistence of catenation after S phase rather depends on cohesin [67]). Interestingly, the hsSMC5/6 complex can accumulate on viral episomal DNA in human hepatocytes [125,126,127,128]. hsSMC5/6 may restrict viral infection by inhibiting transcription of viral genes from the episomal DNA. Mechanism of this restriction is not clear, but it may correlate with a specific topology of the circular episomal DNA and specific composition of the SMC5/6 complex [129].

In higher eukaryotes, SMC5/6 is loaded onto chromatin before or during DNA replication in a manner dependent on the initiation of DNA synthesis, and it dissociates from chromatin during mitosis [130,131]. For example, in *Xenopus laevis* egg extracts, the chromatin binding of xlSMC5/6 is induced by origin firing and elongation [130]. When the egg extracts enter mitosis, xlSMC5/6 gradually dissociates from chromatin (while condensin is loaded). The induction of DSBs following replication does not significantly affect the amount of chromatin-associated xlSmc6, suggesting that the xlSMC5/6 complex is either pre-localized to potential DNA damage sites that may arise during replication or is localized to replication sites to assist in their normal processing (it takes part in DNA damage repair or stabilization of stalled replication forks if necessary). 

Consistent with the (non-repair) chromatin structure (topological) function, depletion of hsSmc5 and hsSmc6 results in aberrant mitotic (curly) chromosome phenotypes that are accompanied by the abnormal distribution of topoisomerase IIα and condensins, and by chromosome segregation errors [131,132]. Curly chromosome phenotypes suggest that the function of hsSMC5/6 is essential to form an intact axial structure of mitotic chromosomes. Significantly, the curly chromosomal phenotypes and the abnormal axial staining of topo IIα, were markedly reduced when cells were subjected to G2 arrest, thereby allowing extra time to complete DNA replication. These data suggest that the hsSMC5/6 function is directly related to a temporal connection between DNA replication and chromosome assembly (also consistent with a role of replication in the shaping of chromosomes and their cohesion [133].

In accordance with the hsSMC5/6 data, inhibition of yeast scSMC5/6 during the S phase causes replication and mitotic delays that are unrelated to aberrant homologous recombination (as *smc6-56* and *smc6-56 rad51∆* mutant cells are equally delayed [121]). In contrast, segregation occurs normally when scSMC5/6 function is inhibited after replication [73,116]. In an *smc6-56 top2-4* double mutant, there is a threefold increase in missegregation, which may indicate that the SMC5/6 complexes recruited upon Top2 inhibition facilitate resolution of SCIs. Altogether, SMC5/6 is intimately involved in the normal progression of replication, most likely as a specific organizer of chromatin structure.

## 4. Conclusions

Despite many efforts to discover the “main function” of SMC5/6 and give the complex an appropriate name, we are still “gathering” data on its multiple functions to obtain a clearer picture. In this review, I summarized only those data that enlightened larger parts of the SMC5/6 puzzle: its role in HR, stabilization of stalled RFs, restart of collapsed RFs, restraining of RF regression, and RF progression in a topological context. Interestingly, all these SMC5/6 functions are related to replication in eukaryotes and may have developed in a very complex way from the role of bacterial SMCs in replication. This assumption is strongly supported by our recent work that suggested striking similarity in the architecture of bacterial SMCs and eukaryotic SMC5/6 complexes [29]. Future investigations and new insights into replication processes in both kingdoms of life will (hopefully) result in a unified model of SMC5/6 function. In addition, a similar architecture of bacterial SMCs and eukaryotic SMC5/6 complexes implies their similar mechanics. It will be interesting to employ new technologies such as cryo-electron microscopy and single-molecule imaging to compare the motor activity of SMC5/6 to the other SMC complexes and find similarities (or differences) between them.

## Figures and Tables

**Figure 1 genes-10-00007-f001:**
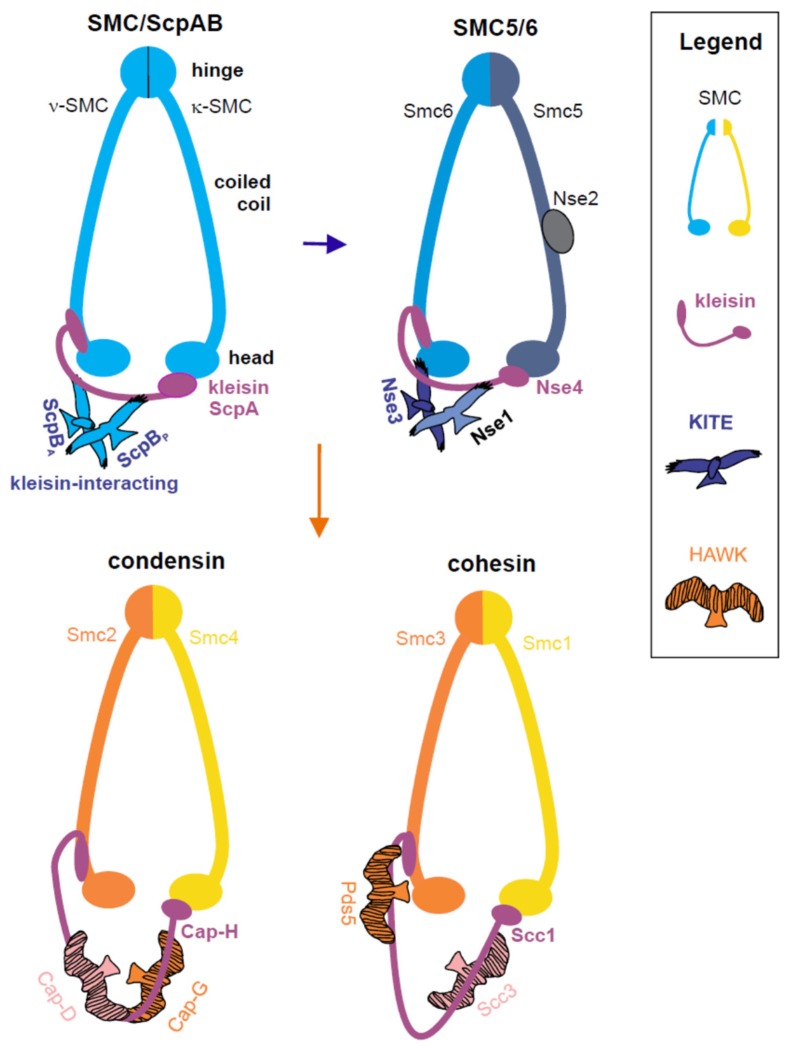
The structural maintenance of chromosome (SMC) complexes form rings. The long coiled-coil SMC proteins interact via their hinge domains at one pole and their heads are bridged at the other pole by the kleisin protein (violet; kleisin’s N- and C-terminal domains bind different SMC proteins). There are two types of kleisin-interacting subunits in different SMC complexes. In the prokaryotic SMC/ ScpAB (segregation and condensation protein) and eukaryotic SMC5/6 complexes, the kleisin molecules bind to the kleisin-interacting tandem winged-helix element (Kite) proteins (suggesting an evolutionary path from prokaryotes to eukaryotes via an ancestral SMC5/6-like complex; blue arrow). In the eukaryotic condensin and cohesin complexes, kleisins interact with the HEAT proteins associated with kleisin (Hawk) proteins (suggesting their later evolution; orange arrow). In addition, the SMC5/6 complex contains a unique conserved Nse2 subunit attached to the SMC5 coiled-coil arm, which possesses small ubiquitin-like modifier (SUMO)-ligase activity. The non-conserved Nse5 and Nse6 subunits are omitted for simplicity.

**Figure 2 genes-10-00007-f002:**
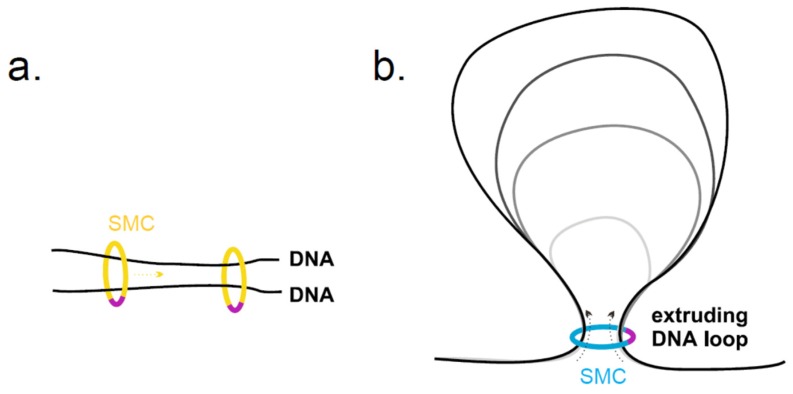
The SMC rings can topologically embrace two DNA strands. The circular shape and the size of the SMC rings enable them to encircle two DNA strands. Two DNA strands can originate from either two sister chromatids (**a**); (cohesin and SMC5/6 can interconnect two DNA strands [66,67]) or two segments of the same DNA molecule (**b**); (most SMC complexes are proposed to form intramolecular loops on DNA). (**a**) ATP binding and hydrolysis provide the SMC complexes with motor activity which allows their translocation along the DNA molecule. (**b**) SMC motor activity may also progressively enlarge loops of DNA in a process termed loop extrusion. Given the loop extrusion activity of the SMC/ScpAB and condensin complexes [64,65], it is very likely that SMC5/6 complexes can also form intramolecular loops on DNA.

**Figure 3 genes-10-00007-f003:**
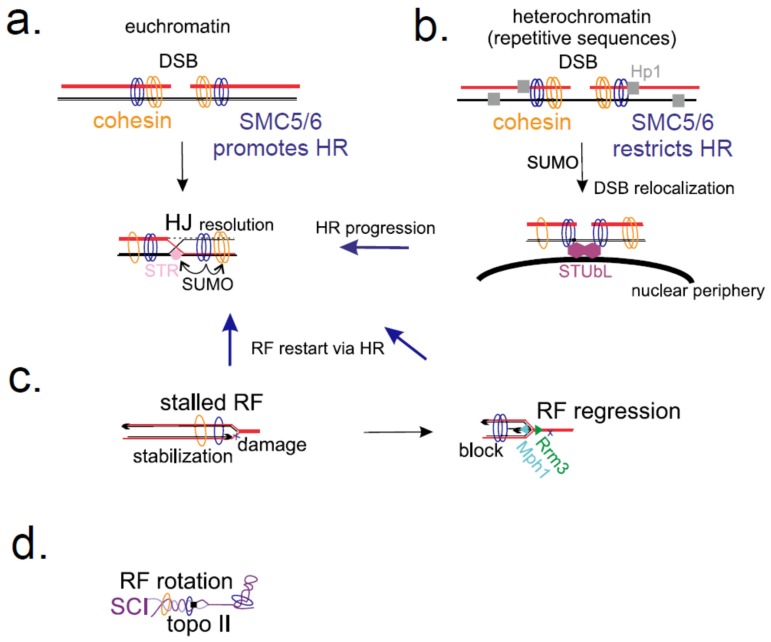
Multiple roles of SMC5/6 in genome integrity maintenance and replication. (**a**) SMC5/6 (blue ring) is recruited to a double-stranded break (DSB) and promotes its repair via homologous recombination (HR) in several ways. It may assist cohesin (yellow ring) in holding sister chromatids and protects it via SUMOylation. In addition, SMC5/6-dependent SUMOylation of STR (Sgs1/Top3/Rmi1; pink circle) helicase promotes resolution of the Holliday junction (HJ). (**b**) SMC5/6 binds heterochromatin protein 1 (HP1; grey square) and blocks HR within the heterochromatin domain to prevent ectopic recombination. Nse2-mediated SUMOylation triggers the relocalization of repair sites to nuclear periphery, where Smc5/6 interacts with the SUMO-targeted ubiquitin-ligase (STUbL) complex (violet hexagon), which ubiquitylates SUMOylated proteins and enables HR progression (blue arrow) outside the heterochromatin domain. (**c**) SMC5/6 localizes to stalled replication fork (RF) and assists in its restart via HR when it collapses (blue arrow). SMC5/6 restricts RF regression (which would otherwise also lead to HR; blue arrow) promoted by Mph1/FANCM helicase (cyan triangle). Pausing sites (such as repetitive ribosomal DNA (rDNA)) are particularly prone to RF regression. The Rrm3 helicase (green triangle) facilitates replication through such pausing sites. (**d**) SMC5/6 promotes RF rotation to relax topological tensions (by topoisomerase II; black square) and remove sister chromatid intertwines (SCI) during replication.

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
