# Peer review of "SMC5/6: Multifunctional Player in Replication"

_genes, 2018, doi:10.3390/genes10010007_

Round 1
Reviewer 1 Report
Comment to ms draft entitled "SMC5/6: multiplayer in replication"
In this review, the author introduces molecular composition and known functions of SMC complexes, including cohesin, condensin, and SMC5/6 complexes. The body of this article consists of brief descriptions on SMC5/6 function, specifically in the context of homologous recombination and DNA replication. I appreciate the author’s endeavor to collect remarkable information from numerous published works. On the other hand, it would be difficult for readers to imagine how SMC5/6 might work by going through somewhat patchy collection of specific observations. I do understand that we are still at the early stages of SMC5/6 study, but I would expect to learn more about perspective views and key questions from the author, as detailed below.
Specific comments:
1. A significant part of manuscript is used to describe all SMC complexes, but I am not sure if this is a good idea, given the limited space for this article. I would keep the introductory part short and save the space for the main topic, namely SMC5/6 and replication.
2. Figure 2 shows how cohesin and condensin associate with DNA. As less is clear for SMC5/6, these figures might be misleading and I would leave them out. Alternatively, the author can cite references to indicate possible DNA binding mode of SMC5/6.
3. A hypothetical model depicting how SMC5/6 and other molecules including helicases might associate with replication fork will be worth adding. This can be done by revising Figure 3d and 3e.
4. The author points out that the SMC5/6 complex is closely related to prokaryotes SMC/SpcAB, more than cohesin or condensin. This is an interesting view. The author may want to speculate any possible function of SMC5/6 based on the function of prokaryotes SMC/SpcAB.
5. To facilitate further studies on SMC5/6, can the author raise what needs to be investigated, and what are the future direction? These perspectives should also help to consider how might SMC5/6 work at the mechanistic level.
Minor points:
6. I find ‘motor activity’ is misleading (line 8), as mechanistically how condensin promote loop extrusion remain unknown. It would be acceptable to call it ‘motor-like activity’.
7. In Figure3 legend, arrow from (c) to (b) should say (b) to (c). Figure legend is missing for (d) and (e). Typo: ‘supress’ to ‘suppress’.
Author Response
Dear reviewer,
Thank you for your comments. They have all been addressed bellow (in red) and in the manuscript (in tracking-changes mode). Furthermore, I added new information and references. You will find new additional references at the end of the manuscript (not inserted to main reference list for technical reasons).
Wish you Merry Christmas and Happy New Year
Jan Palecek
Reviewer 1
In this review, the author introduces molecular composition and known functions of SMC complexes, including cohesin, condensin, and SMC5/6 complexes. The body of this article consists of brief descriptions on SMC5/6 function, specifically in the context of homologous recombination and DNA replication. I appreciate the author’s endeavor to collect remarkable information from numerous published works. On the other hand, it would be difficult for readers to imagine how SMC5/6 might work by going through somewhat patchy collection of specific observations. I do understand that we are still at the early stages of SMC5/6 study, but I would expect to learn more about perspective views and key questions from the author, as detailed below.
Specific comments:
A significant part of manuscript is used to describe all SMC complexes, but I am not sure if this is a good idea, given the limited space for this article. I would keep the introductory part short and save the space for the main topic, namely SMC5/6 and replication.
As the second reviewer asked for more info, I rather tried to improve the Introduction chapter.
Figure 2 shows how cohesin and condensin associate with DNA. As less is clear for SMC5/6, these figures might be misleading and I would leave them out. Alternatively, the author can cite references to indicate possible DNA binding mode of SMC5/6.
Corrected
A hypothetical model depicting how SMC5/6 and other molecules including helicases might associate with replication fork will be worth adding. This can be done by revising Figure 3d and 3e.
Figure 3 was revised.
The author points out that the SMC5/6 complex is closely related to prokaryotes SMC/SpcAB, more than cohesin or condensin. This is an interesting view. The author may want to speculate any possible function of SMC5/6 based on the function of prokaryotes SMC/SpcAB.
Our lab is currently testing a hypothesis based on the SMC5/6-SMC/ScpAB similarity and we do not want to disclose more than what is in the current manuscript version (very sorry).
To facilitate further studies on SMC5/6, can the author raise what needs to be investigated, and what are the future direction? These perspectives should also help to consider how might SMC5/6 work at the mechanistic level.
The Conclusion chapter was extended.
Minor points:
I find ‘motor activity’ is misleading (line 8), as mechanistically how condensin promote loop extrusion remain unknown. It would be acceptable to call it ‘motor-like activity’.
Given the early data published recently, the motor activity of the SMC complexes is just being uncovered and may still cast some doubts about it. However, as the SMC complexes actively move along DNA and extrude loops in ATP-dependent way, I consider them as real motors (as proposed in following papers: [46] Terakawa, Science, 2017; [54] Ganji, Science, 2018; [7] Hassler, Curr Biol, 2018).
In Figure3 legend, arrow from (c) to (b) should say (b) to (c). Figure legend is missing for (d) and (e). Typo: ‘supress’ to ‘suppress’.
Figure 3 was revised.

Reviewer 2 Report
Comments for Palecek Smc5/6 review
This review covers a very interesting and mysterious DNA binding complex called Smc5/6. This review should be accepted pending revisions.
Major Comments:
1. The current title doesn’t quite make sense (Smc5/6: multiplayer in replication). “Multiplayer” would suggest many copies or versions of Smc5/6 playing various roles in replication. I recommend a title such as “Smc5/6: multifunctional in replication”.
2. The author should define the various subunits of both the human and yeast Smc5/6 complexes early on in the review. He should also briefly mention what is known about the role of each component and comment on what it might mean for function that yeast Smc5/6 contains more subunits than human Smc5/6.
3. The author should add information concerning the role of Smc5/6 in inhibiting transcription (and subsequent DNA replication) of circular episomes such as HBV cccDNA. Smc5/6 inhibits the replication of circular episomes such as HBV cccDNA by blocking pgRNA transcription and subsequent reverse-transcription to DNA. Smc5/6 has been shown to localize to PML/Sp100 which are a part of Nuclear Domain 10 bodies (ND10) in non-dividing primary human hepatocytes. This localization is required for Smc5/6 to repress HBV cccDNA transcription. However, in the presence of hepatitis B virus X protein (HBx), Smc5/6 is ubiquitinated by the DDB1-Cul4 E3 ligase and proteosomally degraded (for more information please see the HBx review in Viruses by Livingston et al. 2017 as well as the papers by Decorsiere et al. 2016 Nature, Murphy et al. 2016 Cell Rep, and Niu et al., 2017 PLoS One). In addition, it should be mentioned that the inhibitory effect of Smc5/6 on episomal transcription may be connected to the proposed topological entrapment activity of Smc5/6 (topological entrapment is mentioned on line 203).
Minor Comments:
The English grammar and word usage should be corrected by a native speaker. Some important examples that need to be fixed:
1. “Structure Maintenance of Chromosome” needs to be changed to “Structural Maintenance of Chromosome” (lines 14 and 34 for example).
2. In several cases “constrains” needs to be changed to “constraints” (line 13 is just one example).
3. Line 243 “insides” should be “insights”.
4. Numerous cases where articles (the, a, or an) are needed or should be deleted.
Other comments on the writing:
1. “RF” needs to be defined in the abstract.
2. “ScpAB” should be spelled out and defined (line 36).
Author Response
Dear reviewer,
Thank you for your comments. They have all been addressed bellow (in red) and in the manuscript (in tracking-changes mode). Furthermore, I added new information and references. You will find new additional references at the end of the manuscript (not inserted to main reference list for technical reasons).
Wish you Merry Christmas and Happy New Year
Jan Palecek
Reviewer 2
This review covers a very interesting and mysterious DNA binding complex called Smc5/6. This review should be accepted pending revisions.
Major Comments:
The current title doesn’t quite make sense (Smc5/6: multiplayer in replication). “Multiplayer” would suggest many copies or versions of Smc5/6 playing various roles in replication. I recommend a title such as “Smc5/6: multifunctional in replication”.
The title was changed to “Smc5/6: multifunctional player in replication”
The author should define the various subunits of both the human and yeast Smc5/6 complexes early on in the review. He should also briefly mention what is known about the role of each component and comment on what it might mean for function that yeast Smc5/6 contains more subunits than human Smc5/6.
Information was added (see lines 69-85). Although Nse5 and Nse6 are not conserved, there are present in yeast, plant and human SMC5/6 complexes. Our data (unpublished) suggest, that they are positioned roughly at the same place in human as in S. pombe complexes.
The author should add information concerning the role of Smc5/6 in inhibiting transcription (and subsequent DNA replication) of circular episomes such as HBV cccDNA. Smc5/6 inhibits the replication of circular episomes such as HBV cccDNA by blocking pgRNA transcription and subsequent reverse-transcription to DNA. Smc5/6 has been shown to localize to PML/Sp100 which are a part of Nuclear Domain 10 bodies (ND10) in non-dividing primary human hepatocytes. This localization is required for Smc5/6 to repress HBV cccDNA transcription. However, in the presence of hepatitis B virus X protein (HBx), Smc5/6 is ubiquitinated by the DDB1-Cul4 E3 ligase and proteosomally degraded (for more information please see the HBx review in Viruses by Livingston et al. 2017 as well as the papers by Decorsiere et al. 2016 Nature, Murphy et al. 2016 Cell Rep, and Niu et al., 2017 PLoS One). In addition, it should be mentioned that the inhibitory effect of Smc5/6 on episomal transcription may be connected to the proposed topological entrapment activity of Smc5/6 (topological entrapment is mentioned on line 203).
Information was added (see lines 248-252).
Minor Comments:
The English grammar and word usage should be corrected by a native speaker. Some important examples that need to be fixed:
1. “Structure Maintenance of Chromosome” needs to be changed to “Structural Maintenance of Chromosome” (lines 14 and 34 for example).
Done
2. In several cases “constrains” needs to be changed to “constraints” (line 13 is just one example).
Done
3. Line 243 “insides” should be “insights”.
Done
4. Numerous cases where articles (the, a, or an) are needed or should be deleted.
Manuscript has been corrected by a native English speaker.
Other comments on the writing:
1. “RF” needs to be defined in the abstract.
RF is already defined in the Abstract.
2. “ScpAB” should be spelled out and defined (line 36).
Done
